# Twenty-Four-Year Trends in Family and Regional Disparities in Fruit, Vegetable and Sugar-Sweetened Beverage Consumption among Adolescents in Belgium

**DOI:** 10.3390/ijerph18094408

**Published:** 2021-04-21

**Authors:** Manon Rouche, Maxim Dierckens, Lucille Desbouys, Camille Pedroni, Thérésa Lebacq, Isabelle Godin, Benedicte Deforche, Katia Castetbon

**Affiliations:** 1Research Centre in “Epidemiology, Biostatistics and Clinical Research”, School of Public Health, Université libre de Bruxelles (ULB), 1070 Brussels, Belgium; Lucille.desbouys@ulb.be (L.D.); Camille.pedroni@ulb.be (C.P.); Theresa.lebacq@ulb.be (T.L.); katia.castetbon@ulb.be (K.C.); 2Department of Public Health and Primary Care, Ghent University (UGent), 9000 Ghent, Belgium; maxim.dierckens@ugent.be (M.D.); benedicte.deforche@ugent.be (B.D.); 3Service d’Information Promotion Education Santé (SIPES), School of Public Health, Université libre de Bruxelles (ULB), 1070 Brussels, Belgium; 4Research Centre in “Social Approaches to Health”, School of Public Health, Université libre de Bruxelles (ULB), 1070 Brussels, Belgium; isabelle.godin@ulb.be

**Keywords:** dietary habits, social disparities, trends, adolescents

## Abstract

Dietary habits are influenced by various determinants that may evolve over time. This study aimed to examine, among adolescents in Belgium, trends in the dietary habits between 1990 and 2014 and to determine changes in family and regional disparities related to diet during this time period. In the 1990, 2002 and 2014 cross-sectional “Health Behaviour in School-aged Children” (HBSC) surveys, food consumption was estimated using a short Food Frequency Questionnaire. The Relative Index of Inequality (RII) enabled quantification of the gradients of inequality related to the family structure and to the region for non-daily fruit and vegetable and daily sugar-sweetened beverage (SSB) consumption. Between 1990 and 2014, the prevalence of non-daily fruit consumption increased from 27.7% to 60.6%, whereas the daily SSB consumption decreased from 58.9% to 34.8%. Over time, a downward trend in family disparities (*p* = 0.007) was observed for daily fruit consumption (RII: 1.58 (1.33–1.88) to 1.18 (1.13–1.23)). An upward trend in region-related disparities (p < 0.001) for SSB was found (RII: 1.15 (1.07–1.23) to 1.37 (1.28–1.47)). The overall trend of increasing disparities when dietary habits improved and decreasing disparities when dietary habits worsened highlights the need to implement actions that improve overall dietary habits while ensuring that disparities do not increase.

## 1. Introduction

The dietary habits of adolescents may be determined by a set of individual factors [1], such as the family structure [2], and contextual factors [1], such as the region of a country [3]. Reducing such disparities may be translated into different policy actions, with the combination of these leading to more effective strategies [1].

However, family structure-related disparities in adolescents’ dietary habits have been scarcely documented [4]. Adolescents from single-parent and blended families, i.e., composed of a biological/adoptive parent and a non-biological/adoptive parent, are more likely to have less favorable dietary habits compared to those from two-parent families [2,5]. This may be related to there being fewer financial resources for single parents, reducing their access to healthy food [6]. In addition, in blended families, step-parents may be less involved in the health education of their stepchildren [5].

Moreover, country region-related disparities in dietary habits have been highlighted in several studies [3,7,8]. For instance, in Switzerland, dietary habits in adults differed according to the linguistic region. One hypothesis was the cultural influence of the neighboring countries [8]. Similarly, in Belgium, adolescents in Flanders are more likely to have healthier dietary habits than those of the two other Belgian regions, i.e., Wallonia and Brussels-Capital [3,7]. Among other determinants, such disparities can be explained by the influence of the neighboring countries, especially for Flanders and Wallonia, and by different cultural [9] and economic [10] backgrounds and food supplies.

Furthermore, the societal context has changed and may have impacted the extent of disparities over time. The few studies on trends in dietary disparities mainly focused on beverages and ethnicity in the U.S. and highlighted persistent disparities [11,12]. Between 1989–1991 and 2007–2008, the consumption of fruit-based and soft drinks per capita intake increased along with disparities related to ethnicity among children [11]. However, between 2003–2004 and 2013–2014, ethnic disparities among children and adolescents regarding the typical intake of fruit drinks declined [12]. This example highlights the need to understand the evolution of such disparities. Indeed, public health initiatives should not only aim at improving the diet of the whole population (population-based approach) [13] or of those at the bottom of a social gradient (targeted approach) [14], but they should also aim to reduce disparities (proportionate universalism approach) [15].

For example, the changing societal context has resulted in a shift of the household structure in recent years. A change in marriage and divorce rates has resulted in an increase in single-parent and blended families [16]. This shift may have been accompanied by changes in the role of the step-parent in blended family structures. Additionally, the three administrative Belgian regions, which are federated entities, are Flanders (mainly Dutch-speaking), Wallonia (mainly French-speaking) and Brussels-Capital (both Dutch- and French-speaking). In recent years, health promotion has been decentralized under the responsibility of the regions. This may have resulted in variable exposure to public health messages regarding diet according to the region.

Under these circumstances, we assume that such societal context changes may have influenced dietary habits and related disparities. The aims of this study were twofold: (i) to describe trends in the prevalence of non-daily fruit and vegetable and daily sugar-sweetened beverage (SSB) consumption among adolescents in Belgium between 1990 and 2014 and (ii) to determine how the dietary disparities related to family structure or to school region have evolved during this time period.

This research is one of the first to investigate, in adolescents, trends over a long period of time regarding different social disparities in several food groups with appropriate methods for analyzing disparities over time. Through our two objectives, our findings might help identify key areas to improve current public health actions on both overall food consumption and related disparities.

## 2. Materials and Methods

### 2.1. Study

Since 1983–1984, the World Health Organization collaborative cross-national “Health Behaviour in School-Aged Children” (HBSC) survey has taken place every four years (in almost 50 countries at present) [17]. The survey aims to produce comprehensive indicators supporting the implementation of health promotion policies and interventions. The standardized research protocol has mostly been constant over the years, enabling the analysis of trends.

In Belgium, which is a regionalized country, the HBSC survey has been conducted independently in French-speaking schools since 1986 and in Dutch-speaking schools since 1990. Altogether, these surveys cover the three regions, namely Wallonia, Flanders and Brussels-Capital, with the latter including both French- and Dutch-speaking schools. The present research was based on the surveys conducted in 1990, 2002 and 2014 in French- and Dutch-speaking schools of the three regions.

Following advice from school authorities, no written consent was requested for French-speaking schools; for the Dutch-speaking schools, an opt-out consent process was implemented. For each survey, questionnaires were self-administrated in a classroom. Adolescents were clearly informed on the survey content and on their right to refuse the completion of the entire questionnaire or specific questions. All procedures used during data collection enabled confidentiality and anonymity. 

### 2.2. Sampling

The sampling plans, developed in order to achieve representativeness of the estimators for each linguistic region, were based on a random sample stratified proportionally to the school networks and included public and private schools. In addition, depending on the survey, samples were stratified proportionally to the province and/or the type of education (general, technical, vocational, etc.). For each survey, schools from the mainstream school system were first randomly selected in each stratum based on an official list of all schools in Belgium. The sampling was repeated for each survey, regardless of whether or not some schools had participated in the previous survey(s). Then, one class from the fifth grade of elementary school (corresponding to adolescents aged ±10 years) to the final grade of secondary school (corresponding to adolescents aged ±18 years) was randomly selected in each grade among the schools participating in the study. In Flanders, several classes may have been selected when different types of education were available in the school. All adolescents in the selected classes were invited to participate over a period of approximately two months, regardless of their health status or social characteristics when they attended one of the participating schools.

In 1990, 2002 and 2014, data from 8866, 32,048 and 23,688 questionnaires were collected, respectively (Appendix A). Adolescents aged 20 or over (included only in French-speaking schools) were excluded. Participants with no missing data for all covariates and fruit consumption (i.e., the most frequently filled-in dietary variable) were included in the analyses. The rate of missing data ranged from 1.0% to 3.0%. Thus, the maximum number of adolescents included in the analyses was 8001 in 1990, 29,825 in 2002 and 21,939 in 2014 (Appendix A).

### 2.3. Measures

The HBSC questionnaire is available in two to three versions adapted to the adolescents’ age. With regard to these analyses, the questions were exactly the same for all adolescents, regardless of their age.

A validated short Food Frequency Questionnaire (sFFQ) was used [18]—the most reliable tool that could be used given the conditions of data collection. Adolescents were asked how often they usually consumed 16 food groups in 1990 and 23 in 2002 and 2014. Fruits, vegetables (“cooked” and “raw” separately in 1990) and sugar-sweetened beverages (SSBs) were selected for the analyses. Five-answer categories were proposed in 1990 in the Dutch-speaking questionnaire—“every day”; “every week”; “every month”; “less than once a month”; and “never”—and in the French-speaking questionnaire—“more than once a day”; “once a day”; “at least once a week”; “rarely”; and “never”. In 2002 and 2014, seven-answer categories were proposed in French- and Dutch-speaking schools: “more than once a day”; “once a day”; “5–6 days a week”; “2–4 days a week”; “once a week”; “less than once a week”; and “never”. 

The family structure variable was based on the people declared by the adolescents as living in their main house and, if applicable, in another house. Three categories were determined: “two-parent family” (adolescents living with both parents in the same house); “blended family” (those living with one parent and one step-parent); and “single-parent family” (those living with a single parent). Based on the address of the school, the school regions were (also referred to as) “Flanders” (Flemish), “Wallonia” (Walloon) and “Brussels-Capital” (Brussels).

### 2.4. Statistical Analyses

#### 2.4.1. Reprocessing Data

All food group answers were categorized as “daily” and “non-daily” consumption in order to correspond, as closely as possible, to the Belgian nutritional recommendations [19], while also being determined by the original answer modalities. In all surveys (except for the 1990 Dutch-speaking assessment), daily consumption corresponded to a consumption of “more than once a day” and “once a day”. In the 1990 Dutch-speaking survey, daily consumption corresponded to consumption “every day”. All of the other answer categories corresponded to non-daily consumption. In order to focus on unfavorable consumption, dietary outcome was “non-daily” consumption of fruit and vegetables and “daily” consumption of SSB. 

Presumed inequalities related to the family structure and to the school region were ranked based on prior hypotheses about the overall dietary pattern of adolescents [3,7]. It was found that two-parent families may be considered as more likely to have favorable dietary habits than single-parent families, with blended families in an intermediate position [7]. In fact, single-parent families may have unhealthier dietary habits, partly due to their limited money and time resources [6] impairing their financial access to healthy food and limiting their availability for preparing healthy meals. In addition, step-parents may be less involved in the nutritional education of children in blended families [5]. Accordingly, the ranking considered here was as follows: two-parent family, blended family and single-parent family. Regional disparities are less homogenous and can differ greatly according to the food group [3,7]. Overall, Walloon adolescents tend to have an intermediate position between Flemish and Brussels adolescents [3,7]. Regional differences may be due to a more or less healthy influence of neighboring countries on the regions [7,8] and different food and socioeconomic environments. In addition, Belgian and regional public health actions might have a different effect [3]. Thus, the retained ranking for the school region was Flanders, Wallonia and Brussels-Capital.

Following such a ranking, a modified ridit transformation [20,21] was distinctly applied to the family structure (model A) and to the school region (model B), and for each survey year, in order to obtain a ranking score *x_i_* between 0 and 1. The family and regional ranks *x_i_* were determined as follows for each family and regional category and for each survey year: (1)$$\frac{c_{i}+c_{i-1}}{2}$$
where *c_i_* is the fraction of the population in the class *i* or lower (*c*_0_ = 0 and *c_i_* = 1) [21]. 

Hypothetical best-placed adolescents had a score equal to 0, while hypothetical worst-placed adolescents had a score equal to 1.

#### 2.4.2. Modeling

Following Mackenbach and Kunst [22], the Relative Index of Inequality (RII) and the Slope Index of Inequality (SII) were used to quantify the gradient of inequality related to the family structure (model A) and to the school region (model B) in relative and absolute terms, respectively. The RII and SII are “the expected relative and excess risks comparing the hypothetical extremes of the scale under the log-linear and linear models, respectively, that best approximate the relation” [23] between the family structure or the school region and dietary habits. These measures are complementary, as relative inequalities and absolute inequalities can evolve in opposite ways, especially with high frequency of unfavorable consumption. Together, therefore, they provide a reliable overview of the situation and prevent wrongly concluding that inequalities have been reduced, for instance. Another advantage of these summary measures of inequality is that they take into account “both the population size and the relative […] position of groups” [22].

(Non-)daily food consumption, expressed as prevalence, was assumed to depend on the notional regional and family ranks *x* and was denoted as *f*(*x*) [21]. To estimate *f*(*x*), a generalized linear model was used with a log link function to calculate the RII
(2)$$\left[\frac{f\left(1\right)}{f\left(0\right)}\right]$$
and an identity link function to calculate the SII
(3)[f(1)−f(0)]
where 0 is the “position of the hypothetical best-placed” adolescents and 1 is the “position of the hypothetical worst-placed” adolescents [23]. Given that the odds ratio estimated by logistic regressions could overestimate the prevalence ratios in cross-sectional studies, a Poisson regression with robust variance was used as a generalized linear model [24].

Following their definitions, the RII and SII can be interpreted as a rate ratio and a rate difference, respectively. An RII equal to 1 means no difference in relative inequalities and an SII equal to zero means no difference in absolute inequalities. A 95% confidence interval (95% CI), i.e., α = 0.05, including 1 for the RII and 0 for the SII means that there is no difference in relative and absolute inequalities, respectively. An RII equal to *x* means that worst-placed adolescents are *x*-fold more likely to have a given food frequency consumption than best-placed adolescents. An SII equal to *x* means that worst-placed adolescents are *x* times more likely to have a given food frequency consumption than best-placed adolescents.

For each food group, crude RII and SII values were estimated for models A and B first. Secondly, RII and SII were adjusted for sex, age and school region (model A) or family structure (model B). Trends over time in RII and SII were estimated by including a two-way interaction term modified ridit score (based on family structure for model A and on school region for model B) by survey in the related model equation [25]. In the case where the evolution of RII and SII was quadratic, i.e., an extremum observed in 2002, a test for quadratic trend was performed (including survey year squared in the two-way interaction term).

Due to the substantial sample sizes, the significance of the differences in adolescents’ characteristics was determined by the magnitude of the differences rather than the *p*-value. The significance of the RII and SII was based on the 95% CI, while for trends, the *p*-value of the two-way interaction term was used [25]. All analyses were performed using Stata/IC 14.2^®^ (StataCorp, College Station, TX, USA).

## 3. Results

Characteristics of adolescents were considered to be stable over the years, except for the family structure (Appendix A). “Single-parent” and “blended” families became more common over time. In addition, in 2014, compared to previous survey years, more adolescents from schools in Wallonia and less adolescents from schools in Flanders were included. Between 1990 and 2014, the prevalence of daily SSB consumption decreased, while the overall prevalence of non-daily fruit and vegetable consumption increased, with a slight decrease since 2002 (Appendix A).

### 3.1. Disparities Related to the Family Structure

Over the years, disparities related to the family structure were observed for the three food groups, except for SSB consumption in 1990 (Table 1). For instance, in 1990, hypothetical worst-placed adolescents (i.e., from single-parent families) were 1.58-fold (95% CI: 1.33–1.88) more prone to non-daily fruit consumption than the hypothetical best-placed (i.e., from two-parent families). 

Over time, a downward trend in family structure-related relative disparities (*p* = 0.007) was observed for non-daily fruit consumption. This was related to a differential increase in non-daily consumption according to the family structure between 1990 and 2014 (Figure 1a). Indeed, initially the biggest non-daily consumer group, adolescents from a blended family had the lowest increase in non-daily fruit consumption between 1990 and 2014. Conversely, adolescents from a two-parent family, i.e., the smallest non-daily consumer group in 1990, showed the highest increase in non-daily consumption. Considerable disparities for non-daily vegetable consumption were observed every survey year (Figure 1b); however, no trend (*p* = 0.32) was found. Conversely, a trend towards increasing disparities (*p* < 0.001) was observed for daily SSB consumption due to an increase in consumption differences between adolescents from different family structures (Figure 1c). Indeed, while they already had the lowest daily consumption, adolescents from a two-parent family experienced the largest decrease in daily SSB consumption. The smallest decrease in daily SSB consumption was observed among adolescents from a single-parent family.

### 3.2. Disparities Related to the School Region

Regional disparities were observed for all food groups over the years, except for SSB consumption in 2002 and for vegetable consumption in 2014 (Table 2). In 2014, hypothetically worst-placed adolescents (i.e., in a school in Brussels) were 0.55-fold (95% CI: 0.52–0.57) more likely to have non-daily fruit consumption than hypothetically best-placed adolescents (i.e., in a Flemish school). 

Throughout the years, a trend towards increasing disparities with time (*p* < 0.001) was observed for non-daily fruit consumption. This was related to a differential change in non-daily consumers according to the school region (Figure 2a). The lowest increase in non-daily fruit consumption was observed among adolescents from Walloon schools. Those from Flemish schools had the highest increase in consumption, although they were the largest consumer group in 1990. Interestingly, adolescents considered as better-placed, i.e., adolescents in a Flemish school, were more likely to have non-daily fruit consumption (RII < 1).

In addition, a trend towards decreasing regional disparities (*p* < 0.001) for non-daily vegetable consumption was found: the lowest increase in non-daily vegetable consumption was observed among adolescents in a Walloon school, while the highest increase was among those in a Flemish school (Figure 2b). Conversely, an upward trend (*p* < 0.001) was observed for daily SSB consumption. This increase was mainly due to the highest decrease among the initially smallest daily SSB consumer group, i.e., adolescents in a Flemish school, and the lowest decrease among the initially largest daily consumer group, i.e., adolescents in a Brussels-Capital school (Figure 2c).

## 4. Discussion

Throughout the years, the overall prevalence of non-daily fruit and vegetable consumption increased (with a slight decrease since 2002), while family and regional disparities decreased, except for regional disparities for non-daily fruit consumption. In contrast, the overall prevalence of daily SSB consumption decreased, whereas family and regional disparities increased. The major changes in food consumption were mainly observed between 1990 and 2002. This is most likely due to the characteristics of the adolescents and their environment rather than to methodological issues (as discussed below).

To the best of our knowledge, few studies have investigated trends in dietary habits, especially in adolescent populations [12,26,27,28] and covering such a long period (>20 years) [28], thus limiting their comparability. The prevalence of non-daily fruit and vegetable consumption doubled between 1990 and 2002 and slightly decreased thereafter, in line with the literature, which highlighted a similar trend for fruit consumption [27]. In addition, the HBSC reports (starting in 1993–1994) confirmed an increase in non-daily fruit and vegetable consumers between the 1990s and early 2000s [29,30,31]. Not eating fruit and vegetables regularly may be due to the unpleasant taste perceived by some adolescents compared with more palatable foods and to practical constraints such as washing, peeling or cooking before consumption compared with ready-to-eat snacks [32]. Fruit, frequently eaten as a snack/dessert, could be replaced by more attractive foods [32], such as biscuits or milky desserts. Furthermore, adolescents may have progressively replaced fruit and vegetables with various industrial processed ready-to-eat fruit-based or vegetable-based foods, such as applesauce or composite dishes. Adolescents may not identify such foods as a consumption of fruit or vegetables and may consequently underreport their consumption when filling out a short FFQ, such as in the HBSC survey. Between 2002 and 2014, the observed small decrease in non-daily fruit and vegetable consumers could partly be explained by public health efforts regarding such foods resulting in a possible increased awareness of the general population on their benefits [33].

The proportion of daily SSB consumers decreased by almost half between 1990 and 2014, with most of the decrease being observed between 1990 and 2002. The observed decrease in daily SSB consumers is in line with most of the HBSC countries since 1998 [30,31,34] (information was not available in the 1993–1994 international HBSC report). Previous studies in the US [12,26] have also highlighted a decline in SSB intake since 1990. For years, policy stakeholders and health professionals have advised the public to limit intake of sugary drinks due to their association with various negative health outcomes [35]. In addition, this warning has potentially pushed industries to progressively market artificially sweetened beverages (ASBs). Although the potential adverse effects of these drinks on health have recently been underlined [36], ASBs were suggested as substitutes for sugary drinks due to their low calorie content. As a result of both the awareness on SSBs and the development of ASBs, adolescents could have substituted SSBs for ASBs [37], which would explain the decrease in SSB consumption between the 1990s and the early 2000s. However, this hypothesis cannot be verified with our data, as the question regarding ASBs was not included until 2002 in the Dutch-speaking schools and 2006 in the French-speaking schools.

To our knowledge, no study has analyzed trends in family and regional disparities regarding dietary habits, thus limiting comparison. Nonetheless, to understand the reasons behind the changes and to provide tools to implement public health initiatives, potential causes of disparities are discussed.

Single-parent families, often with fewer financial resources (one-third had a low family affluence scale (FAS) [38] in 2002 and 2014 compared to one in six for other family structures), may have adopted the new dietary recommendations more slowly than other family structures did, including the limitation of SSB consumption. Indeed, income is associated with adherence to nutritional recommendations [39]. In addition, these families may have been less likely to substitute SSBs for ASBs due to the slightly higher average price of the latter [40]. Moreover, in recent years, blended families have become increasingly frequent at the expense of two-parent families. In Belgium, the share of blended families was less than 5% in 1990, while it was almost 10% in 2002 and 15% in 2014. Since step-parents are usually less active and involved in the education and the health of the children [5], we assumed that the increase in the blended family structure could have led to a shift in the role of step-parents, resulting, in turn, in a change in the families’ dietary habits. In addition, adolescents from blended families reported a higher FAS in 2014 compared to 2002 (84.3% had a medium or a high FAS in 2014 vs. 79.3% in 2002). Therefore, healthy foods, such as fruits and vegetables, may have become more affordable [6] over the years for a greater proportion of blended families. 

In recent years, in Belgium, the change in federated entities responsible for health promotion led to different campaigns and interventions being implemented in the regions, in terms of messages, target population, dissemination or intensity. In addition, the effectiveness of these actions may be different given that the responsiveness could be culturally influenced [11]. Our results suggest that public health initiatives focusing on fruit and vegetable consumption were less effective in the Flemish population than they were in other regions, while those for SSB consumption were more effective in Flanders. In addition, we assumed that the food supply and the culture-related behaviors, which are complex to evaluate, may have changed differently over the years depending on the region, but this requires further examination. Furthermore, the regional changes in food consumption were also similar to the changes observed in neighboring countries, i.e., in France between 1993–1994 and 2013–2014 [29,30,31] and in the Netherlands between 2001–2002 and 2013–2014 [30,31]. Indeed, a higher increase in non-daily fruit consumption and a higher decrease in daily SSB consumption were observed in the Netherlands compared to France, confirming the influence of neighboring countries. However, no similarity for vegetable consumption was observed with neighboring countries [30,31]. The imbalanced increase in non-daily vegetable consumers resulted in a homogenization of behaviors in Belgium.

A major strength of our study was the long retrospective period (24 years), with repeated and similar assessments of dietary habits. This enabled to highlight trends that would have been different compared to a shorter timer period that would have only started in the 2000s. However, some characteristics could not be used because they were not comparable across the surveys or were not yet developed, such as the Family Affluence Scale [38]. In addition, the food groups available for analyses since 1990 were rather limited, but the selected groups are amongst the most important in terms of health [35] and have particularly been subject to public health actions. The long-term assessment led to a potential bias related to the small differences between the questionnaires. Indeed, the number of response categories increased from five to seven, and some foods were grouped together (“raw” and “cooked” vegetables into one category) or specified (sugar- and non-sugar-sweetened beverages were more clearly differentiated). These changes should not cause a major methodological issue [27] and, therefore, are unlikely to be responsible for the major changes occurring since the 1990s. Nevertheless, they may be responsible for a slight overestimation of the consumption of vegetables in 1990, for instance. The initial response categories also prevented a more accurate categorization of food consumption in relation to the dietary recommendations. Furthermore, the large sample size led to a gain in statistical power but did not hinder the confidence in tests for trends based on interaction terms. The methodology used to evaluate the disparities has the advantage of considering whole regional/family distribution and not only the extreme groups as is the case in classic comparisons. Indeed, social inequalities cannot be reduced on the extremes of the social hierarchy. Actions, especially those directed at high-risk populations, could shift the risk to intermediate groups. Therefore, meaningful information on the changes in disparities would have been missed with classic measures. Furthermore, thanks to the *x*-rankings of the family and the regional distributions, these indices allow valid cross-population comparisons [23]. However, although the changing structure of the population is taken into account, their disadvantage is that they do not actually capture a possible change in the structure of the population. These measures must, therefore, be complemented by the population attributable factor. Indeed, the change in disparities was a genuine change and was not due to changes in the population structure (data not shown). Finally, in order to facilitate the interpretation of the findings, the same category order based on prior assumption of the overall dietary pattern [3,7] was kept throughout the analyses. However, depending on the year and the food group studied, the ranking of family structure and school region categories may be somewhat different. Given the small differences observed, keeping the original order only led to a slightly lower accuracy of these estimates.

## 5. Conclusions

In summary, increasing disparities were observed when dietary habits improved, whereas decreasing disparities were observed when dietary habits worsened. This trend is to be confirmed with other food groups, e.g., whole grains or animal protein foods such as dairy or meat, although the consistency of our results for the three food groups suggests that the same trend will be observed. Furthermore, changes in other significant determinants of dietary habits, such as socioeconomic or migration status, must be studied when available. 

While it was suspected and hypothesized, this is the first time that such a figure has been demonstrated for several indicators and food groups and over a long period of time. Our results underline that public health actions in Belgium have, thus far, failed to both improve dietary habits and tackle social inequalities, possibly due to a counterproductive environment such as intensive marketing of unhealthy foods. Furthermore, our study confirms that interventions regarding dietary habits must better mobilize the concept of proportionate universalism, i.e., improving dietary habits proportionally to the degree of needs. More practically, actions should take into account the specificities, needs and access barriers to a healthy diet of different sub-populations in order to be universal, but with an intensity that depends on the level of disadvantage. Thus, improved affordability of healthy foods and including a cultural component in actions could help improve the dietary habits of adolescents while reducing family- and regional-related inequalities, respectively.

## Figures and Tables

**Figure 1 ijerph-18-04408-f001:**
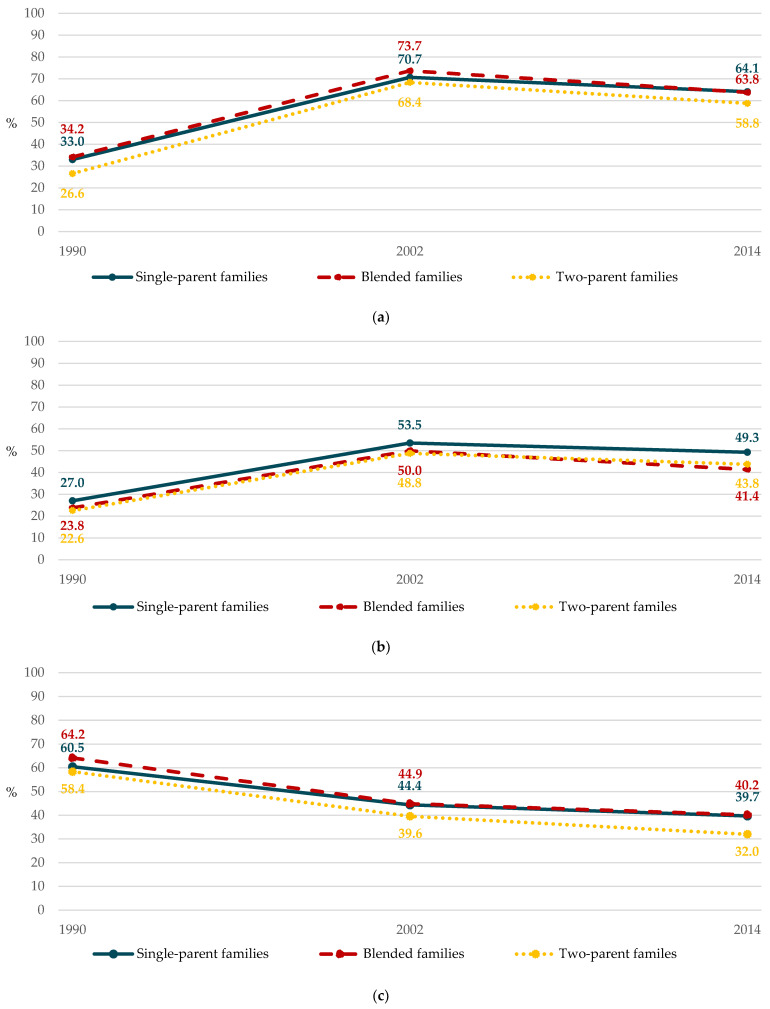
Changes in prevalence of food consumption according to the family structure between 1990 and 2014 (HBSC, Belgium, 1990–2002–2014): (**a**) non-daily fruit; (**b**) non-daily vegetable; (**c**) daily sugar-sweetened beverage.

**Figure 2 ijerph-18-04408-f002:**
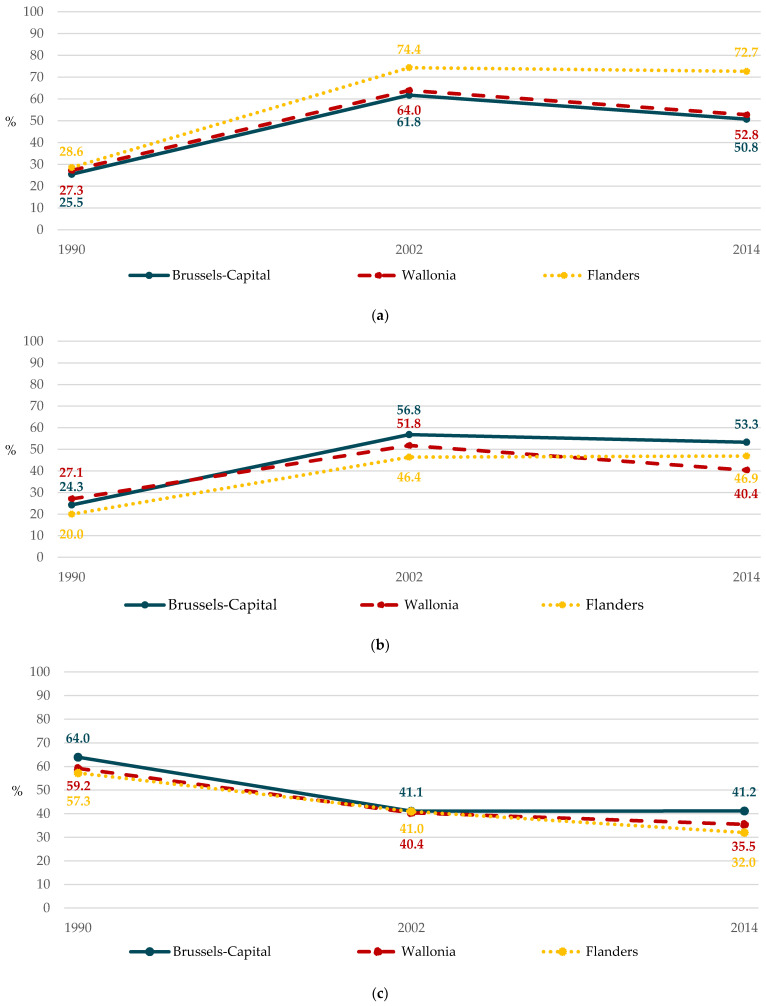
Changes in prevalence of food consumption according to the school region between 1990 and 2014 (HBSC, Belgium, 1990–2002–2014): (**a**) non-daily fruit; (**b**) non-daily vegetable; (**c**) daily sugar-sweetened beverage.

**Table 1 ijerph-18-04408-t001:** Prevalence and Relative Index and Slope Index of Family Inequalities (RII-F and SII-F) of fruit, vegetable and sugar-sweetened beverage consumption among adolescents between 1990 and 2014 (HBSC, Belgium, 1990–2002–2014).

Family Structure	1990	2002	2014	*P* for Trend
**Non-daily fruit consumption**				
Single-parent family (%)	33.0	70.7	64.1	
Blended family (%)	34.2	73.7	63.8	
Two parents (%)	26.6	68.4	58.8	
cRII-F (95% CI) ^1^	1.56 (1.32–1.86)	1.10 (1.06–1.14)	1.17 (1.13–1.23)	0.43 ^a^
aRII-F (95% CI) ^2^	1.58 (1.33–1.88)	1.15 (1.11–1.19)	1.18 (1.13–1.23)	0.007 ^a^
cSII-F (95% CI) ^1^	13.37 (7.82–18.92)	6.56 (4.10–9.02)	9.91 (7.26–12.55)	<0.001 ^b^
aSII-F (95% CI) ^2^	13.14 (7.59–18.70)	9.53 (7.04–12.03)	10.40 (7.72–13.08)	<0.001 ^b^
**Non-daily vegetable consumption**				
Single-parent family (%)	27.0	53.5	49.3	
Blended family (%)	23.8	50.0	41.4	
Two parents (%)	22.6	48.8	43.8	
cRII-F (95% CI) ^1^	1.35 (1.10–1.66)	1.14 (1.09–1.21)	1.15 (1.08–1.22)	0.26 ^a^
aRII-F (95% CI) ^2^	1.30 (1.06–1.60)	1.11 (1.06–1.17)	1.16 (1.09–1.23)	0.32 ^a^
cSII-F (95% CI) ^1^	7.30 (2.08–12.51)	6.80 (4.11–9.50)	6.10 (3.42–8.78)	0.68 ^a^
aSII-F (95% CI) ^2^	6.74 (1.54–11.94)	5.56 (2.84–8.28)	6.81 (4.11–9.50)	0.76 ^a^
**Daily sugar-sweetened beverage consumption**				
Single-parent family (%)	60.5	44.4	39.7	
Blended family (%)	64.2	44.9	40.2	
Two parents (%)	58.4	39.6	32.0	
cRII-F (95% CI) ^1^	1.10 (1.00–1.21)	1.26 (1.18–1.34)	1.51 (1.41–1.62)	<0.001 ^a^
aRII-F (95% CI) ^2^	1.08 (0.98–1.19)	1.27 (1.19–1.35)	1.47 (1.37–1.58)	<0.001 ^a^
cSII-F (95% CI) ^1^	5.87 (0.00–11.74)	9.70 (7.00–12.40)	15.05 (12.39–17.72)	0.001 ^a^
aSII-F (95% CI) ^2^	4.80 (−1.11–10.70)	10.10 (7.41–12.80)	14.34 (11.68–17.00)	0.002 ^a^

^1^ Crude Relative Index of family Inequality and Slope Index of family Inequality and their 95% confidence intervals; ^2^ Adjusted Relative Index of family Inequality and Slope Index of family Inequality for sex, age and school region, and their 95% confidence intervals; ^a^
*P* for linear trends; ^b^
*P* for quadratic trends.

**Table 2 ijerph-18-04408-t002:** Prevalence and Relative Index and Slope Index of regional Inequalities (RII-R and SII-R) of fruit, vegetable and sugar-sweetened beverage consumption among adolescents between 1990 and 2014 (HBSC, Belgium, 1990–2002–2014).

School Region	1990	2002	2014	*P* for Trend
**Non-daily fruit consumption**				
Brussels-Capital (%)	25.5	61.8	50.8	
Wallonia (%)	27.3	64.0	52.8	
Flanders (%)	28.6	74.4	72.7	
cRII-F (95% CI) ^1^	0.87 (0.76–0.99)	0.74 (0.71–0.76)	0.54 (0.52–0.57)	<0.001 ^a^
aRII-F (95% CI) ^2^	0.83 (0.73–0.96)	0.73 (0.71–0.76)	0.55 (0.52–0.57)	<0.001 ^a^
cSII-F (95% CI) ^1^	−3.92 (−7.67–−0.17)	−20.75 (−22.76–−18.73)	−34.84 (−37.15–−32.53)	<0.001 ^a^
aSII-F (95% CI) ^2^	−4.19 (−7.94–−0.45)	−20.71 (−22.76–−18.67)	−34.68 (−37.03–−32.33)	<0.001 ^a^
**Non-daily vegetable consumption**				
Brussels-Capital (%)	24.3	56.8	53.3	
Wallonia (%)	27.1	51.8	40.4	
Flanders (%)	20.0	46.4	46.9	
cRII-F (95% CI) ^1^	1.57 (1.35–1.82)	1.31 (1.25–1.37)	0.97 (0.91–1.02)	<0.001 ^a^
aRII-F (95% CI) ^2^	1.55 (1.33–1.80)	1.29 (1.23–1.34)	0.99 (0.93–1.05)	<0.001 ^a^
cSII-F (95% CI) ^1^	11.07 (7.39–14.76)	13.43 (11.24–15.62)	−1.44 (−3.79–0.91)	<0.001 ^a^
aSII-F (95% CI) ^2^	11.38 (7.65–15.10)	12.65 (10.44–14.87)	−0.38 (−2.75–1.98)	<0.001 ^a^
**Daily sugar-sweetened beverage consumption**				
Brussels-Capital (%)	64.0	41.1	41.2	
Wallonia (%)	59.2	40.4	35.5	
Flanders (%)	57.3	41.0	32.0	
cRII-F (95% CI) ^1^	1.13 (1.06–1.22)	0.99 (0.94–1.04)	1.35 (1.26–1.45)	<0.001 ^b^
aRII-F (95% CI) ^2^	1.15 (1.07–1.23)	0.99 (0.94–1.04)	1.37 (1.28–1.47)	<0.001 ^b^
cSII-F (95% CI) ^1^	7.41 (3.32–11.49)	−0.55 (−2.72–1.62)	10.27 (7.89–12.66)	<0.001 ^b^
aSII-F (95% CI) ^2^	8.13 (4.00–12.26)	1.13 (−1.05–3.31)	11.03 (8.65–13.40)	<0.001 ^b^

^1^ Crude Relative Index of regional Inequality and Slope Index of regional Inequality and their 95% confidence intervals; ^2^ Adjusted Relative Index of regional Inequality and Slope Index of regional Inequality for sex, age and family structure and their 95% confidence intervals; ^a^
*P* for linear trends; ^b^
*P* for quadratic trends.

## Data Availability

The data presented in this study are available on request from the corresponding author.

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
