# Peer review of "Twenty-Four-Year Trends in Family and Regional Disparities in Fruit, Vegetable and Sugar-Sweetened Beverage Consumption among Adolescents in Belgium"

_ijerph, 2021, doi:10.3390/ijerph18094408_

Round 1

Reviewer 1 Report

The authors present a study to examine, among adolescents in Belgium, trends in the dietary habits during twenty four years, and to determine changes in family and regional disparities related to diet during this time period.

The article is correctly written and presented, although the results section is somewhat messy to read.

The major strength of the study is the long observation period (24 years), however the article provides little new in terms of its conclusion; the conclusion they get about the need for to implement actions that improve overall dietary habits while ensuring that disparities do not increase, it's expected and not very new. Nor is it anything new their conclusion about the fact that interventions regarding dietary habits must better mobilize the concept of proportionate universalism, i.e., improving dietary habits proportionally to the degree of needs.

I fully agree with the authors when they state, in the Discussion, that “we assumed that the food supply and the culture-related behaviors, which are complex to evaluate, may have changed differently over the years depending on the region” and that “this requires further examination” changed differently over the years depending on the region, but this requires further examination”. Indeed, that is one of the keys.

Regarding the study methodology, I am aware of the difficulty of a nutritional study in large population groups, since I have participated in some, but the abbreviated FFQ provides too limited information and even more so if there is not a trained nutritionist carrying out the survey.

Minor:

  • SSB in the abstract must be defined (sugar-sweetned beverage) before abbreviating it
  • the article has some typographical errors
  • in the bibliography some citations include the DOI and others do not

Author Response

Dear Reviewer

We would like to thank you for your relevant comments and suggestions to improve our manuscript. We have revised our manuscript according to all  comments (changes highlighted in yellow).  Please find below our answers to each comment and the corresponding revisions in the text.

The authors present a study to examine, among adolescents in Belgium, trends in the dietary habits during twenty four years, and to determine changes in family and regional disparities related to diet during this time period.

The article is correctly written and presented, although the results section is somewhat messy to read.

We have made the results easier to read and modified the manuscript pages 5-6 lines 244-249, 252-257, page 8 lines 279-283 and 290-293.

The major strength of the study is the long observation period (24 years), however the article provides little new in terms of its conclusion; the conclusion they get about the need for to implement actions that improve overall dietary habits while ensuring that disparities do not increase, it's expected and not very new. Nor is it anything new their conclusion about the fact that interventions regarding dietary habits must better mobilize the concept of proportionate universalism, i.e., improving dietary habits proportionally to the degree of needs.

Though they generally are acknowledged, very few data have actually documented such long-term trends in the “real life” to support such hypotheses i.e. improved dietary habits along with increased inequalities, and reciprocally. Our data also highlight that the need for proportional universalism is still present and important despite various pleas for a long time.

Regarding this issue, we have revised the conclusion page 12 lines 431-435.

I fully agree with the authors when they state, in the Discussion, that “we assumed that the food supply and the culture-related behaviors, which are complex to evaluate, may have changed differently over the years depending on the region” and that “this requires further examination” changed differently over the years depending on the region, but this requires further examination”. Indeed, that is one of the keys.

We are grateful to the reviewer for his/her positive comment.

Regarding the study methodology, I am aware of the difficulty of a nutritional study in large population groups, since I have participated in some, but the abbreviated FFQ provides too limited information and even more so if there is not a trained nutritionist carrying out the survey.

In the HBSC multi-thematic surveys, the tool chosen to describe the dietary habits of adolescents is the short Food Frequency Questionnaire (sFFQ), as it is easier to implement in large-scale and school-based survey. The sFFQ list consists consisted of about ten gross food groups. Albeit its inherent limitations, such a tool has considerable advantages given the conditions of the data collection and the target population. For instance, adolescents can answer the sFFQ without the help of a trained interviewer like a nutritionist. We acknowledged the limitations of such tools, which were discussed and did not overstate our conclusions given such limits. However, we have complemented the manuscript page 3 lines 131-132.

Minor:

SSB in the abstract must be defined (sugar-sweetned beverage) before abbreviating it

We have modified the abstract accordingly page 1 line 22 (the abbreviation was spelled out in the introduction, we have not modified it).

the article has some typographical errors

We have double-checked the manuscript and corrected where necessary.

in the bibliography some citations include the DOI and others do not

We have added all the DOI to follow the instructions.

Reviewer 2 Report

Authors of the manuscript entitled „Twenty-four-year trends in family and regional disparities in dietary habits among adolescents” present an interesting analysis of data on fruit and vegetable consumption trends in adolescents in Belgium, but several points need to be completed and clarified.

Title

In the title, it should be clarified that Authors analyzed trends in the consumption of vegetables and fruits, and that this applies to Belgian adolescents.

Abstract

SSB - abbreviation needs to be expanded.

Materials and Methods

- Did the parents consent to the completion of the FFQ by children?

- Did all children, regardless of age, complete the same questionnaire?

- Questionnaires for younger children require additional explanations and the parent's help in completing them.

- What were the inclusion and exclusion criteria from this survey? Did only healthy children take part in this survey? Did only children with normal body weight take part in this study?

- Did Authors also analyze trends in the consumption of other food groups? Why did Authors focus solely on the consumption of vegetables and fruits? It seems that the analysis of only one food category is a serious drawback of this manuscript, because other groups are also important, e.g. dairy products, cereal products.

- How long did the survey take? Did adolescents complete the FFQ at the same time in all schools? Was the consumption of vegetables and fruit assessed depending on the season (summer / winter)?

- Were the same schools always involved in the study? Were different schools drawn each time?

Results

- Table 1 is in the manuscript twice.

Discussion

- Categorizing fruit and vegetable consumption as daily or non-daily consumption is too simplified considering that the daily consumption category can answer both once a day and several times a day or more, which is not recommended. How will Authors comment obtained results in relation to dietary recommendations, which have also changed over the years? In addition, please indicate the ratio between the consumption of vegetables and fruit.

- Line 279-280 „Lack of regular consumption of  fruit and vegetables may be due to unpleasant taste and inconvenience” – sentence incomprehensible, please correct it.

- Authors incorrectly write about a long observation period, taking into account that Authors compared the results from three time points (1990, 2002 and 2014), it is not possible to write about long-term observation of trends, especially that the questions in the FFQ covered 1 month.

- Please provide practical conclusions for the results. How can they be used in a practical way? Do  Authors believe that nutritional education for one-parent families should be different? Please explain.

References

References must be corrected.

- Authors must complete the DOI for their references    

- Authors should follow the instructions for authors (the way of referring is incorrect).

Author Response

Dear Reviewer

We would like to thank you for your relevant comments and suggestions to improve our manuscript. We have revised our manuscript according to all  comments (changes highlighted in yellow).  Please find below our answers to each comment and the corresponding revisions in the text.

Authors of the manuscript entitled „Twenty-four-year trends in family and regional disparities in dietary habits among adolescents” present an interesting analysis of data on fruit and vegetable consumption trends in adolescents in Belgium, but several points need to be completed and clarified.

Title 

In the title, it should be clarified that Authors analyzed trends in the consumption of vegetables and fruits, and that this applies to Belgian adolescents.

We have modified the title accordingly page 1 lines 2-3.

Abstract

SSB - abbreviation needs to be expanded.

We have modified the abstract accordingly page 1 line 22.

Materials and Methods

- Did the parents consent to the completion of the FFQ by children?

The consent was for the whole survey and not for a specific item such as the sFFQ. Therefore, parents did not give specific consent for the completion of the sFFQ by their children. In addition, adolescents had the right to refuse to answer to any specific questions, including the sFFQ.

We have complemented the manuscript page 3 lines 98-100.

- Did all children, regardless of age, complete the same questionnaire?

For each survey round, there were three questionnaire versions adapted to the age of the adolescents. For example, questions about sex life and illicit drugs are only asked to older adolescents. The sFFQ and family structure questions, as well as gender and age, were addressed in exactly the same way to all adolescents, regardless of their age.

We have complemented the manuscript accordingly page 3 lines 128-130.

- Questionnaires for younger children require additional explanations and the parent's help in completing them.

As mentioned in the answer above, the questionnaires are age-appropriate. In addition, the questions have been pre-tested with adolescents of all ages to ensure understanding. The questions used in our analyses were similar for all adolescents, regardless of their age (see modification page 3 lines 128-130). In general, the adolescents did not need any additional help or explanation to be able, as the words chosen were very simple (see page 3 lines 142-143). Similarly, the use of a sFFQ makes it easier for them to understand the items listed (gross food groups). Finally, the region was collected on the basis of school addresses (see page 3 lines 146-148), thus not involving adolescents in the responses.

We have not modified the manuscript except for the clarification about the questions’ similarity across age groups.

- What were the inclusion and exclusion criteria from this survey? Did only healthy children take part in this survey? Did only children with normal body weight take part in this study?

All adolescents attending a mainstream school in Belgium could be included in the survey. Children who were enrolled in special education, i.e., children with important disabilities, and those who were not attending schools were excluded from the outset. Thus, both healthy and unhealthy children, normal-, under- or overweight but who were in mainstream school system were eligible. In the same way, no exclusion for analysis was made on the basis of health status or any other characteristic of the adolescent. Only age 20 years or older or missing data for covariates or food consumption were responsible for an exclusion, i.e., maximum 3% (see page 3 lines 122-124).

We have complemented the manuscript page 3 lines 109-110 and 117-119.

- Did Authors also analyze trends in the consumption of other food groups? Why did Authors focus solely on the consumption of vegetables and fruits? It seems that the analysis of only one food category is a serious drawback of this manuscript, because other groups are also important, e.g. dairy products, cereal products.

Sugar-sweetened beverage consumption was also analysed (see Tables 1 and 2). The long period of time does not allow for a wide choice of food groups to be studied (see pages 11-12 lines 392-394). In fact, food groups to be selected for analyses have to be asked in each survey round to ensure comparison, i.e., in 1990, 2002 and 2014, but only fruit, vegetables and SSB were available in common. Nonetheless, we have analysed the trends for the three food groups that are amongst the most important in terms of health and are particularly subject to public health actions.

As mentioned in our conclusion (pages 12 lines 425-427), our results should be confirmed with other food groups. The consistency within our findings makes us believe that similar results can be hypothesized, i.e., improving/worsening consumption with increasing/decreasing disparities, respectively. We fully agree that this will deserve additional assessment.

We have complemented the manuscript page 12 lines 394-395 and 427-428.

- How long did the survey take? Did adolescents complete the FFQ at the same time in all schools? Was the consumption of vegetables and fruit assessed depending on the season (summer / winter)?

The data collection generally took place for around two months. In view of this short period of time, it is not expected to have missed a seasonal effect when comparing the different survey years. No separate answers were provided for seasons, they were asked to answer “in general”.

Following this comment, we have complemented the manuscript page 3 lines 117-118 and lines 131-132.

- Were the same schools always involved in the study? Were different schools drawn each time?

All schools can be drawn at random in each survey. In other words, if a school participated in a survey, it was not excluded from the sample for the next survey. Therefore, some included schools for a survey round could be included in the next round thought it was infrequent given the sampling rate.

We have complemented the manuscript regarding this point page 3 lines 111-112.

Results 

- Table 1 is in the manuscript twice.

We have replaced the second table 1 by table 2, corresponding to the disparities related to the school region.

Discussion

- Categorizing fruit and vegetable consumption as daily or non-daily consumption is too simplified considering that the daily consumption category can answer both once a day and several times a day or more, which is not recommended. How will Authors comment obtained results in relation to dietary recommendations, which have also changed over the years? In addition, please indicate the ratio between the consumption of vegetables and fruit.

As indicated on page 3 lines 151-153, the objective of the categorisation was to correspond as much as possible to the dietary recommendations while being limited by the original response modalities. In 1990, the sFFQ had only five response options. In Flanders, adolescents were asked to report a “every day” consumption or a lower frequency (four options). It was not possible to distinguish between “once a day” and “several times a day or more”. Therefore, as explained, the possibilities for categorisation were limited. A categorisation that distinguished between “once a day” and “several times a day or more” might indeed have been closer to the recommendations.

Our answer modalities, i.e. frequencies, did not make possible to study trends in consumption in relation to dietary recommendations, expressed in quantity. However, it would indeed be interesting to study the trend in the proportion of adolescents by social position respecting the dietary recommendations with appropriate data. Moreover, a daily intake of fruit and vegetables has always been recommended, but the quantity to consume per day has evolved. Therefore, the interpretation of our results does not depend of the evolution in dietary recommendations.

We have complemented the manuscript page 12 lines 402-404.

In addition, please indicate the ratio between the consumption of vegetables and fruit.

With these frequency-based classifications, it is unfortunately not possible to obtain a ratio indicator that would be interpretable. Therefore, we have not modified the manuscript.

- Line 279-280 „Lack of regular consumption of  fruit and vegetables may be due to unpleasant taste and inconvenience” – sentence incomprehensible, please correct it.

We have made the sentence clearer page 10 lines 323-326.

- Authors incorrectly write about a long observation period, taking into account that Authors compared the results from three time points (1990, 2002 and 2014), it is not possible to write about long-term observation of trends, especially that the questions in the FFQ covered 1 month.

Following this relevant comment, we have modified the corresponding sentence page 11 line 388.

- Please provide practical conclusions for the results. How can they be used in a practical way? Do  Authors believe that nutritional education for one-parent families should be different? Please explain.

To improve dietary habits and reduce related inequalities, actions must take into account the specificities, needs and access barriers to a healthy diet of the different sub-populations. According to our results, they should include a socioeconomic and sociocultural component. The program stakes do not have to be necessary different, but they do have to meet the criteria above in order not to exclude the single-parent families from the action. The action could be intensified for this sub-population through, for example, an easier access and more support to the nutrition education programs.

We have complemented the manuscript accordingly page 12 lines 438-443.

References

References must be corrected.

References have been corrected when necessary.

- Authors must complete the DOI for their references    

DOIs have been added.

- Authors should follow the instructions for authors (the way of referring is incorrect).

The instructions have been re-read and the referencing has been corrected where necessary.

Reviewer 3 Report

The authors investigate trends in non-daily consumption of fruit, non-daily consumption of vegetables and daily consumption of SSB in Belgium. They monitor whether consumption inequalities of these healthy foods and unhealthy drink have increased or decreased in relation to family structure and region between 1990 and 2014.

I find the topic interesting and timely. Figures are informative and clearly presented. Even if I think the study methods are valid and reliable, I also think that its explanation lacks enough detail to understand them fully. I would suggest stating clearly the definition of the SII and RII and the difference between the two. What do each measure? It is not clear from the current version of the paper. In addition, all the hypotheses in which the family structure and the region categories have been ranked should be clearly referenced (lines 143-151).

In general, I think the authors could exploit the results to a greater extent by giving further detail in a clearer and more orderly manner. In this way, they could enhance a lot the paper.

Table 1 has been included twice in the paper and Table 2 has not been included. Therefore, it is difficult to assess the results and discussion related to the school region.

I would suggest including a short note on the contribution of the paper to the literature at the end of the introduction.

Finally, I miss some discussion on the appropriateness of using the SII and RII in this analysis.

Author Response

Dear Reviewer,

We would like to thank you for your relevant comments and suggestions to improve our manuscript. We have revised our manuscript according to all comments (changes highlighted in yellow).  Please find below our answers to each comment and the corresponding revisions in the text.

The authors investigate trends in non-daily consumption of fruit, non-daily consumption of vegetables and daily consumption of SSB in Belgium. They monitor whether consumption inequalities of these healthy foods and unhealthy drink have increased or decreased in relation to family structure and region between 1990 and 2014.

I find the topic interesting and timely. Figures are informative and clearly presented. Even if I think the study methods are valid and reliable, I also think that its explanation lacks enough detail to understand them fully. I would suggest stating clearly the definition of the SII and RII and the difference between the two. What do each measure? It is not clear from the current version of the paper.

The Relative and Slope Index of Inequality are summary measures of the association between social position and dietary habits. In other words, the RII and SII are “the expected relative and excess risks comparing the hypothetical extremes of the scale under the log-linear and linear models, respectively, that best approximate the relation” between social position and dietary habits (Moreno-Betancour et al. 2015). In summary, the RII is a relative measure, while the SII is an absolute one.

The RII and SII are complementary in the results interpretation. Indeed, all relative differences being equal, an absolute difference can be much higher when frequency of unfavourable food consumption is high compared with a situation where the frequency is low. Furthermore, absolute and relative differences can move in opposite directions, meaning that the absolute difference may decrease over the time while relative difference increases. In this case, studying only relative or absolute inequalities would not give a full picture of the overall situation. Nonetheless, in our analyses, the RII and SII evolved in a similar way, with an increase in SII when RII increased and a decrease in SII when RII decreased.

We have complemented the manuscript page 4 lines 189-196.

Moreno-Betancur, M.; Latouche, A.; Menvielle, G.; Kunst, A.E.; Rey, G. Relative index of inequality and slope index of inequality: a structured regression framework for estimation. Epidemiology 2015, 26, 518–527, doi:10.1097/EDE.0000000000000311

In addition, all the hypotheses in which the family structure and the region categories have been ranked should be clearly referenced (lines 143-151).

We have complemented the manuscript accordingly page 4 lines 161-162, 164-177 and 171-174.

In general, I think the authors could exploit the results to a greater extent by giving further detail in a clearer and more orderly manner. In this way, they could enhance a lot the paper.

We modified the manuscript page pages 5-6 lines 244-249, 252-257, page 8 lines 279-283 and 290-293.

Table 1 has been included twice in the paper and Table 2 has not been included. Therefore, it is difficult to assess the results and discussion related to the school region.

Table 2 has been included page 8.

I would suggest including a short note on the contribution of the paper to the literature at the end of the introduction.

Our research is, to the best of our knowledge, the first to investigate dietary habits and related disparities over such a long period of time. Our findings will help better understand dietary disparities and further, implement efficient public health actions.

We have complemented our manuscript page 2 lines 79-83.

Finally, I miss some discussion on the appropriateness of using the SII and RII in this analysis.

Among other interests, the RII and SII find their relevance in taking into account the entire population structure and in validly comparing cross-populations. Related results will therefore be much more useful than results based on classical measures. The latter not only fail to take into account the changes in the population structure, but also only consider the extremes of the social hierarchy.

We have complemented the manuscript accordingly page 12 lines 408-417.

Round 2

Reviewer 1 Report

I thank the authors for the effort made to improve the manuscript, accepting suggestions

Reviewer 2 Report

Authors revised the manuscript according to comments.

Reviewer 3 Report

The paper has improved. I have no more comments. If anything, I would suggest checking the fact that regional disparities for non-daily fruit consumption have increased. I am not sure it has been correctly interpreted.